# Water Status and Predictive Models of Moisture Content during Drying of Soybean Dregs Based on LF-NMR

**DOI:** 10.3390/molecules27144421

**Published:** 2022-07-10

**Authors:** Tianyou Chen, Wenyu Zhang, Yuxin Liu, Yuqiu Song, Liyan Wu, Cuihong Liu, Tieliang Wang

**Affiliations:** 1College of Engineering, Shenyang Agricultural University, Shenyang 110866, China; chentianyou93@163.com (T.C.); 2021240012@stu.syau.edu.cn (W.Z.); 2020240024@stu.syau.edu.cn (Y.L.); songyuqiusyau@sina.com (Y.S.); wly78528@163.com (L.W.); cuihongliu77@163.com (C.L.); 2College of Water Conservancy, Shenyang Agricultural University, Shenyang 110866, China

**Keywords:** by-products, drying processing, relaxation peak, water status, moisture content prediction

## Abstract

To explore the drying characteristics of soybean dregs and a nondestructive moisture content test method, in this study, soybean dregs were dried with hot air (80 °C), the moisture content was measured using the drying method, water status was analyzed using low-field nuclear magnetic resonance (LF-NMR) and the moisture content prediction models were built and validated. The results revealed that the moisture contents of the soybean dregs were 0.57 and 0.01 g/g(w.b.), respectively, after drying for 5 and 7 h. The effective moisture diffusivity increased with the decrease in moisture content; it ranged from 5.27 × 10^−9^ to 6.96 × 10^−8^ m^2^·s^−1^. Soybean dregs contained bound water (*T*_21_), immobilized water (*T*_22_) and free water (*T*_23_ and *T*_23_’). With the proceeding of drying, all of the relaxation peaks shifted left until a new peak (*T*_23_’) appeared; then, the structure of soybean dregs changed, and the relaxation peaks reformed, and the peak shifted left again. The peak area may predict the moisture content of soybean dregs, and the gray values of images predict the moisture contents mainly composed of free water or immobilized water. The results may provide a reference for drying of soybean dregs and a new moisture detection method.

## 1. Introduction

Soybean dregs are by-products from the processing of tofu, milk, sufu and other soy products [1,2,3]. Approximately 1.4 billion tons are produced per year [4]. The soybean dregs are rich in dietary fibers, proteins, lipids, vitamins and minerals and are of high nutritional values. Intake of soybean dregs may improve blood lipids, prevent obesity and relieve diabetes mellitus [5,6,7]. However, fresh soybean dregs are high in water content, which may induce microbe growth and increase the putrid progress. Therefore, dehydration is important for the utilization of soybean dregs.

Several drying technologies have been investigated to process soybean dregs. The continuous entrained bed dryer [8] and the impinging stream dryer are of high efficiency in drying soybean dregs (okara) [9]. Unfortunately, the hot air temperature used in the continuous entrained bed dryer and the super-heated steam in the impinging dryer were as high as 190 °C, which destroyed the dried okara. Wang et al. [10] employed a response surface methodology and a synthetic evaluation to optimize the drying process of okara in an air jet impingement dryer, and the parameters for temperature, air velocity and loading density were obtained. Hot air drying is the most common method for its simple operation and low investment and operating costs [11,12]. Chen et al. [13] explored the moisture content changing patterns of soybean dregs during hot air drying, and the mathematical model was established. However, the water state and effective water diffusivity were not investigated for the drying of soybean dregs, and they are important for controlling the drying process [14].

Several techniques are available for the determination of moisture content including the drying method, differential scanning calorimetry (DSC), dynamic thermal mechanical analysis (DMTA), near-infrared spectroscopy (NIR) and nuclear magnetic resonance (NMR) methods. However, the operation process of DSC, DMTA and NIR is more complicated and cannot accurately reflect the spatial distribution and binding status of water [15]. The low-field nuclear magnetic resonance (LF-NMR) technique can determine the relaxation of hydrogen protons to investigate the water mobility and distribution of materials [16], and the magnetic resonance images (MRIs) may show the internal distribution of water during the drying process [17,18]. Therefore, LF-NMR can be used to predict moisture content through a peak’s area [19,20], but the mathematical relationship between the moisture content of materials and the gray values of a proton density image has not been investigated.

In this study, the characteristics of soybean dregs during hot air drying was investigated, and LF-NMR was used to analyze the water status change during soybean dregs drying. The mathematical relationships between the moisture content of soybean dregs and transverse relaxation peak areas and between the moisture content of soybean dregs and proton density image gray values were established to provide a reference for the drying of soybean dregs and a nondestructive moisture content testing method.

## 2. Materials and Methods

### 2.1. Materials and Instruments

Fresh soybean dregs with an initial moisture content of 0.7880 g/g(w.b.) were bought from a tofu processing plant, and the variety of soybean was Liaodou 50. 

The weight of the materials was measured with an electronic balance (BS200S, Sartorius, Gottingen, Germany); a hot air drying oven (101, Yongguangming Medical Instruments Co., Ltd., Beijing, China) was used to dry the dregs, and another drying oven (HY-1B, Tongli Xinda Instruments Co., Ltd., Tianjin, China) was used for the complete drying of the samples during the drying process to obtain the dry matter. A nuclear magnetic resonance instrument (MiNiMR-60, Niumag Electronic Technology Co., Ltd., Shanghai, China) was used to obtain the water state and images of the samples during the drying process, which operates at a main magnetic intensity of 0.5 ± 0.05 T, radiofrequency pulse frequency of 12 MHz, corresponding to resonance frequency of 20 MHz, magnet temperature at 32 °C and probe coil diameter of 15 mm.

### 2.2. Methods

#### 2.2.1. Sampling and Treatment

Studies show that high temperatures may destroy the nutrients of soybean dregs, and duration of drying at 80 °C to reduce the moisture content of fresh dregs to a safe storage level was two-thirds less than that of drying at 40 °C [13]. Therefore, the fresh soybean dregs were dried with hot air at 80 °C in this study. The dregs were put on a pan (320 × 300 × 28 mm) and dried in the oven. It was agitated at an interval of 15 min during the drying process to obtain samples with more uniform moisture contents. The sampling was carried out at drying for 1, 2, 3, 4, 5, 5.5, 6, 6.5, and 7 h, since preliminary experiments showed that the water contents of the soybean dregs declined more rapidly at the late stage of drying. The samples and the original materials were, respectively, subjected to hot air drying for water content testing and LF-NMR treatment to obtain the spectral signal of the water state and an image of the hydrogen proton density at the corresponding water content. The testing process is shown in Figure 1. Each test was repeated 3 times.

#### 2.2.2. Detection of Moisture Content

Wet basis (w.b.) moisture content was used in this study, since the signal strength and total mass can be measured directly using LF-NMR treatment [21,22]. The testing of water content of soybean dregs was conducted according to the Detection of Water Content in Foods (Standard GB5009.3-2016) [23]. Samples of 2–5 g were put in a drying oven at 105 °C for 2–4 h; then, they were taken out of the oven, sealed and cooled in a desiccator for 0.5 h and then weighed. Next, the samples were put again into the oven at 105 °C for 1 h, then taking them out of the oven, they were sealed and cooled in the desiccator for 0.5 h and then weighed. The above procedure was repeated until the mass difference was less than 2 mg, and the weight was determined as the mass of the dry matter of the sample. The water content can be calculated using Equation (1).
(1)$$MC=\frac{m-m_{d}}{m}$$
where *MC* is the moisture content(w.b.) in the sample, g/g; *m* is the mass of the sample, g; *m_d_* is the dry mass the sample, g.

#### 2.2.3. Determination of the Effective Moisture Diffusivity

Fick’s second law of diffusion equation was used to fit the experimental drying data for the determination of the effective moisture diffusivity coefficients [24,25].
(2)$$\frac{\partial \left(MR\right)}{\partial t}=D_{eff}\cdot \frac{\partial^{2}\left(MR\right)}{\partial x}$$
where *D_eff_* is the effective moisture diffusivity, m^2^·s^−1^; *t* is the drying time, s; *x* is the spatial displacement of mass transport, m; and *MR* is the moisture ratio, expressed as the moisture content at time *t* divided by initial moisture content, *X_t_*/*X_o_* (dry basis).

The solution by Crank for this partial differential equation for one-dimensional mass transport in infinite slab geometry is given as Equation (3) [26], provided that the drying temperature and migration by diffusion are constant and the shrinkage is negligible.
(3)$$MR=\frac{8}{\pi^{2}}\sum_{n=0}^{\infty }\frac{1}{{\left(2n+1\right)}^{2}}\cdot \exp\left(-\frac{{\left(2n+1\right)}^{2}\pi^{2}D_{eff}t}{4L^{2}}\right)$$
where *L* is the sample thickness, m; *n* is a positive integer.

For long drying periods, the equation can be simplified to the first term of the series [27]. Therefore, assuming that the effective diffusivity does not depend on the moisture content of the sample, then taking natural logarithms on both sides of the equation, Equation (4) is obtained.
(4)$$\ln\left(MR\right)=\ln\left(\frac{8}{\pi^{2}}\right)-\left(\frac{\pi^{2}F_{0}}{4}\right)$$
*F*_0_ = (*D_eff_* · *t*)/*L*^2^ and *F*_0_ is called the dimensionless Fourier number.

Therefore, the effective moisture diffusivity may be calculated as Equation (5).
(5)$$D_{eff}=\frac{F_{0}}{\left(t/L^{2}\right)}$$

#### 2.2.4. Water Detection with LF-NMR

The spectrum of the water in the soybean dregs was tested with LF-NMR. The transverse relaxation time (*T*_2_) of the spectrum of each sample was measured using the pulse Carr–Purcell–Meiboom–Gill (CPMG) sequences. The parameters of the CPMG sequences were dominant frequency: SF = 21 MHz; 90° and 180° radiofrequency pulse widths of *P*1 = 15 μs and *P*2 = 36 μs; waiting time of repeated sampling, *TW* = 1000 ms; number of signal sampling points, *TD* = 90,016; echo number, NECH = 3000; echo time, *TE* = 0.15 ms; repeated sampling number, NS = 64. Inversion was conducted through the iterative reconstruction algorithm: the parameters were relaxation time *T*_2_; the initiation time was 0.01 ms; the number of iterations was 100,000; the cut-off time was 10,000 ms; the number of inversion points was 200.

The cross-sections of the samples were scanned using a multilayer spin echo pulse sequence of LF-NMR, and images of water in soybean dregs were obtained. The main parameters of the magnetic resonance imaging (MRI) were dominant frequency SF01 = 21.65 MHz; 90° and 180° radiofrequency pulse widths: *RAF*90 = 1.0 and *RAF*180 = 2.0; repetition time, *TR* = 300 ms; echo time, *TE* = 5.885 ms; layer number = 3; layer thickness = 3 mm; interlayer space = 1.7 mm.

The relaxation spectral data and relaxation peak areas inverted were mass normalized. The images of soybean dregs dried for different time periods were unified and mapped using LF-NMR image processing software, and proton density gray images were obtained. Five interested regions from these images were randomly selected, and the gray values were averaged. Then, the densities were normalized to obtain the normalized gray value of each sample.

#### 2.2.5. Prediction Model and Error Analysis

Moisture content prediction models were established using a fitting method, and the statistical criteria, such as coefficient of determination (*R*^2^), the residual sum of squares (*RSS*), root mean square error (*RSME*), and reduced chi-square (*χ*^2^), were utilized to obtain the quality of the fitting [28,29]. These statistical criteria can be calculated as:(6)$$R^{2}=1-\frac{\sum_{i=1}^{N}{\left(m_{i}-m_{pi}\right)}^{2}}{\sum_{i=1}^{N}{\left(m_{i}-m_{mi}\right)}^{2}}$$
(7)$$RSS=\sum_{i=1}^{N}{\left(m_{i}-m_{pi}\right)}^{2}$$
(8)$$RMSE=\sqrt{\frac{1}{N}\sum_{i=1}^{N}{\left(m_{i}-m_{pi}\right)}^{2}}$$
(9)$$\chi^{2}=\frac{\sum_{i=1}^{N}{\left(m_{i}-m_{pi}\right)}^{2}}{N-z}$$
where *m_i_* is the experimental value; *m_pi_* is the predicated value; *m_mi_* is the average of experimental values; *N* is the number of observations taken; *z* is the number of constants in the model.

The samples dried for 2.5, 4.5 or 6.6 h were used to validate the moisture prediction models. The moisture content were measured using the drying method. The area of relaxation peak and the gray values of unified mapping proton density were determined from the spectra and proton density images based on LF-NMR. The mean values of relaxation peak area per unit mass and the gray value per unit density were determined through mass and density normalization. The mean values were substituted into predicting models respectively to obtain the predicted moisture content. Then, the prediction errors of the moisture content could be calculated using to equation [30].
(10)$$P_{e}=\frac{\left|y_{m}-y_{p}\right|}{y_{m}}\times 100\%$$
where *P**_e_* is the percent error, %; *y**_m_* is the measured moisture content(w.b.) of soybean dregs, g/g; *y**_p_* is the predicted moisture content(w.b.) of soybean dregs, g/g.

#### 2.2.6. Data Processing

The Origin software (8.0 version, Origin Lab, Northampton, MA, USA) was employed to plot diagrams and build regression models. The SPSS software (26.0 version, SPSS Inc., Chicago, IL, USA) was used to analyze variance by one-way ANOVA and to determine the significance of differences between mean values using Duncan’s multiple range test (*p* < 0.05). The results are presented as the mean ± SD.

## 3. Results and Discussion

### 3.1. Drying Characteristics of Soybean Dregs

#### 3.1.1. Moisture Content

The characteristic curve of hot air drying of soybean dregs is illustrated in Figure 2. The decreasing rate of water content (slope) firstly increased and reached maximum drying during 6–6.5 h, and then there was a small decrease. The moisture contents of soybean dregs were 0.57 and 0.01 g/g(w.b.), respectively, after drying for 5 and 7 h. For most materials, the water loss rate is low in the late drying period [31,32]. The results of Cheng et al. [13] showed that the drying rate of soybean dregs was still relatively high in the late stage of hot air drying. In the microwave drying process of carrot slices, the water loss slowed down but not significantly in the late drying stage [33]. Both investigations obtained similar results to that of this study.

#### 3.1.2. Effective Moisture Diffusivity

The variation of ln(*MR*) with the drying time of soybean dregs is shown in Figure 3, and it is not linear. As a consequence, the effective moisture diffusivity of the soybean dregs will depend on the moisture content as is usually the case for highly porous materials [34]. Celma et al. [35] and Erenturk et al. [25] reported similar trends in the variation of moisture diffusivity for industrial tomato by-products and rosehips, respectively.

The dependence of soybean dregs’ effective diffusivity on moisture content is shown in Figure 4. The value of the effective diffusivity was observed to decrease with the increase of moisture content. The effective moisture diffusivity of soybean dregs was 5.27 × 10^−9^–6.96 × 10^−8^ m^2^·s^−1^ when the moisture content was 0.72–0.01 g/g(w.b.). Celma et al. [35] reported a similar trend in the variation of moisture diffusivity for industrial tomato by-products, and the range of effective moisture diffusivities was 5.179 × 10^−9^ m^2^/s–1.429 × 10^−8^ m^2^/s.

### 3.2. Water Status during Drying of Soybean Dregs

The soybean dregs dried for different times were scanned with LF-NMR to obtain the inverted spectra of transverse relaxation time *T*_2_, as shown in Figure 5. According to the combination of the strength of water and matter, water in material is generally divided into bond water, immobile water and free water [36,37]. It can be observed from Figure 5, that there are four transverse relaxation peaks and corresponding times (i.e., *T*_21_, *T*_22_, *T*_23_ and *T*_23_’) of soybean dregs on LF-NMR spectra. The first peak (*T*_21_) represents the bound water, which is closely attached to the cell wall components [38]. The second peak (*T*_22_) indicates immobile water fixed within highly organized structures [39]. Free water can more easily absorb magnetic energy to vibrate compared with the other two forms of water, and it needs longer time to return to the ground state after the disappearance of the magnetic field. The relaxation times of the third peak (*T*_23_) and the fourth peak (*T*_23_’) were long, which are the relaxation peak of free water [40]. The relaxation peaks of different forms of water in the soybean dregs were slightly different from those in other agricultural products, since the bonding forces of water differ among different forms of water. The water in soybean dregs evaporated with the proceeding of drying; thus, the moisture content declined, and the relaxation peaks of the spectra of the soybean dregs dropped significantly, indicating that the size of a relaxation peak reflects the moisture content during the drying of soybean dregs.

It is reported that the contents of free water, immobilized water and bound water in a material are positively correlated with the corresponding transverse relaxation peak areas [41]. To clarify the water status in the soybean dregs during the drying process, the time of the relaxation peaks and the areas of the relaxation peaks per unit mass corresponding to the free water, immobilized water and bound water of the samples were analyzed, as shown in Figure 6.

The time of relaxation peaks during drying of soybean dregs is shown in Table 1 and the comparative analysis of the water status changes during drying is shown in Figure 6. The dominant form of water in soybean dregs was free water during drying for 0–4 h. The amounts of free water and immobilized water decreased with time, and the quantity of bound water decreased first and then increased. After the water with weak bonding force in the soybean dregs evaporated, the bonding forces of the remaining water were relatively enhanced, and the three relaxation peaks shifted left. Sun et al. [23] and Chitrakar et al. [40] reported that the structural changes in the internal material caused by drying released carbohydrates and other components, which resulted in the reduction in water fluidity and shortening of the relaxation time, and the water combination in the material became increasingly tight.

After drying for approximately 4 h, the marginal free water with weak bonding force generated a new relaxation peak. The surface of the soybean dregs contracted and deformed after drying for approximately 5 h, the free water turned into immobilized water and immobilized water became bound water, resulting in a decrease in free water and increases in immobilized water and bound water. As a result, immobilized water became dominant, and new relaxation peaks formed.

During drying for 5–6 h, the moisture content decreased rapidly, the bonding force of the remaining water became strong and the relaxation peak shifted left. During this period, some of the immobilized water became bound water because of the change in the structure, and some immobile water transferred to the surface of the soybean dregs because of being heated, became free water and then evaporated. Qu et al. [42] reported that weak binding water is converted into free water with time and then evaporates during the drying process of potato and oat composite noodles. Li et al. [12] and Zhang et al. [43] also reported that free water of *Panax quinquefolium* L. and *Arctium lappa* L. increased at the late stage of hot air drying.

After drying for approximately 6.5 h, the internal contraction and deformation of soybean dregs occurred, new relaxation peaks appeared, a large amount of immobilized water became bound water or free water and, thus, bound water became the major existing form of water.

During drying for 6.5–7 h, the moisture content decreased, and the bonding force of the remaining water was enhanced and the relaxation peaks shifted left, but the bound force in some of the bound water weakened when it was heated, and the bound water became free water. As a result, the content of the free water increased, and the major existing form of water became free water.

### 3.3. Construction and Validation of Moisture Content Prediction Model

#### 3.3.1. Relationship between LF-NMR Peak Area and Moisture Content

Comparison of the LF-NMR transverse relaxation peak areas between the fresh soybean dregs and the soybean dregs dried over different times showed that the changes in the areas were mainly caused by the moisture content of the soybean dregs. Hence, the relationship between the mass normalized peak areas and the moisture content was established; the fitting curve is shown in Figure 7. The moisture content of the soybean dregs is obviously linearly related to the peak area per unit mass. A liner model was set-up with a fitting method based on the experimental results of the moisture content and peak area per unit mass as shown as Equation (11). The fitting *R*^2^ was 0.98; *RSS* was 0.03; *RSME* was 0.03; *χ**^2^* was 0.001, indicating that the regressed straight line well fit the observed values.
(11)$$y=1.42\times 10^{-5}x_{1}-0.14$$
where *y* is the moisture content(w.b.) of the soybean dregs, g/g; *x*_1_ is the total peak area per unit mass on the LF-NMR spectra (*T*_21_ + *T*_22_ + *T*_23_ +*T*_23_’).

#### 3.3.2. Relationship between the Gray Value of Proton Density and the Moisture Content

The above LF-NMR transverse relaxation tests showed that the LF-NMR signals mainly originated from the hydrogen protons of internal water of the soybean dregs. MRI can also reflect the moisture distribution in soybean dregs. Hence, the pseudo-color images of the proton density of the soybean dregs dried over different times were obtained through imaging and unified mapping with LF-NMR, as shown in Figure 8. The layers were set for scanning with LF-NMR, and each layer was 3 mm thick. The first, second and third layers are numbered from top to the bottom to indicate the layer of the samples.The red (high brightness) stands for large hydrogen proton density and high moisture content, and the blue (low brightness) presents a small hydrogen proton density and low moisture content [44].

The pseudo-color values of the proton density images decreased significantly with the increase in drying time, indicating the decline in the hydrogen proton density and moisture content of the soybean dregs. Hence, the images can directly reflect the changes in the moisture content in the soybean dregs. However, the pseudo-color values in the late stage of drying declined rapidly, and the values of the hydrogen proton images after drying for 6.5 h (moisture content: 16.31 g/g, w.b.) and 7 h (moisture content: 1.19 g/g, w.b.) were approximately the same. The reason for this phenomenon is that after drying for 5–6 h, the water in soybean dregs mainly existed as immobilized water, and the relaxation peak of the immobilized water shifted left with the drying time, resulting in a decrease in the immobilized water involved in imaging. After drying for 6 h, the immobilized water involved in imaging decreases. The water in soybean dregs dried for 6.5 h mainly existed as bound water, and its relaxation peak time was less than the echo time (*TE*) of the MRI and, thus, this part of the water was not involved in the imaging.

The soybean dregs dried for 0–5.5 h in which most of the hydrogen protons of the water molecules were involved in imaging were sampled, and the gray values from the unified mapped images were taken and density normalized to explore the relationship between the moisture content and the gray value of the hydrogen proton density image; the data are shown in Figure 9. It can be concluded that the experimental results of the moisture content and the gray value per unit density were linearly related, the fitted liner model was set-up and is shown as Equation (12). The fitting *R*^2^ of 0.93, *RSS* of 0.02, *RSME* of 0.03, and *χ*^2^ of 0.0009, indicate that the regressed straight line well fit the observed values.
(12)$$y=3.97\times 10^{-3}x_{2}+0.02$$
where *y* is the moisture content(w.b.) of soybean dregs, g/g; *x*_2_ is the gray value of the hydrogen proton density image per unit density.

#### 3.3.3. Prediction Error Analysis

The measured moisture content and predicted moisture content of soybean dregs when dried for 2.5, 4.5 or 6.6 h were obtained through testing and prediction Equations (11) and (12). Then, the prediction errors of the moisture content were determined using Equation (10), the results are listed in Table 2. It can be observed that the prediction errors of the moisture content were 1.69–8.58% based on the peak areas, and the prediction errors were 1.39–3.48% based on the gray values. The prediction errors in both cases were less than 10%, which satisfies the requirement in the Chinese National Standard for Water Measuring Methods for Materials (Standard GB 5009.3-2016).

In conclusion, the prediction model based on LF-NMR can precisely predict the moisture content of soybean dregs, and the prediction model based on MRI gray values can mainly predict the contents of free water or immobilized water in soybean dregs.

## 4. Conclusions

In this study, the change in the moisture content and effective moisture diffusivity of soybean dregs while drying were experimentally determined. The decreasing rate of the water content (slope) firstly increased and reached its maximum during 6–6.5 h, and then decreased slightly. The effective moisture diffusivity increased as the moisture content reduced. The water status of soybean dregs during drying were studied using LF-NMR, and the results showed that the different forms of water transformed into each other, and the relaxation peaks shifted and reformed. The linear models between moisture content and peak areas and gray values were established and verified, and the prediction errors were less than 10%. However, the gray value cannot be used to predict the moisture content of materials mainly composed of bound water due to the short relaxation time.

## Figures and Tables

**Figure 1 molecules-27-04421-f001:**
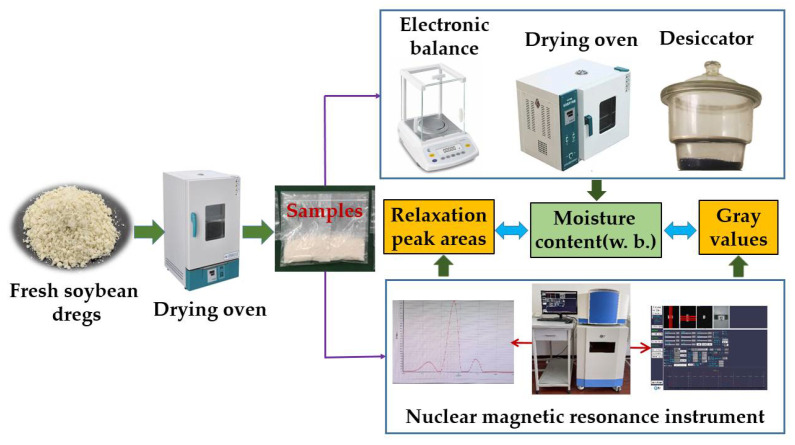
Drying process and moisture measurement of soybean dregs.

**Figure 2 molecules-27-04421-f002:**
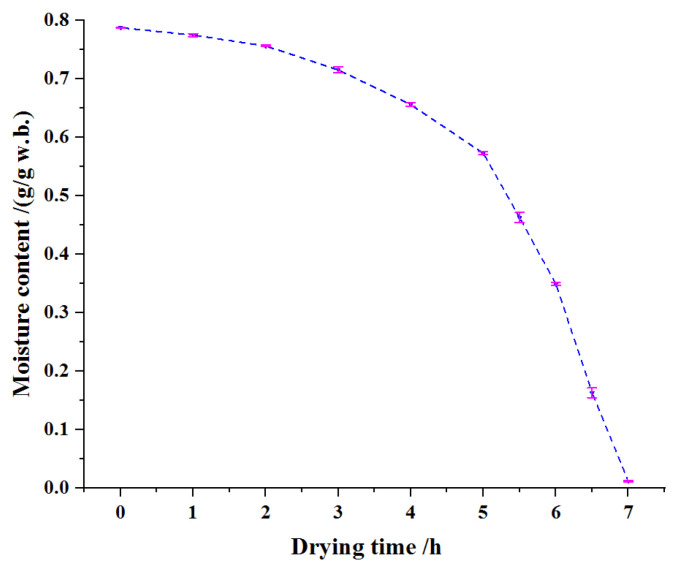
Drying curve of soybean dregs.

**Figure 3 molecules-27-04421-f003:**
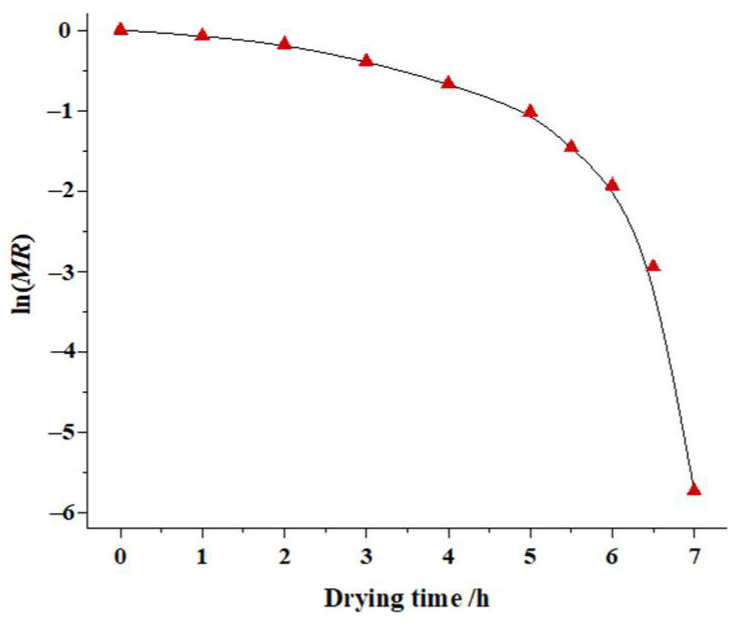
Plot of ln (*MR*) versus drying time.

**Figure 4 molecules-27-04421-f004:**
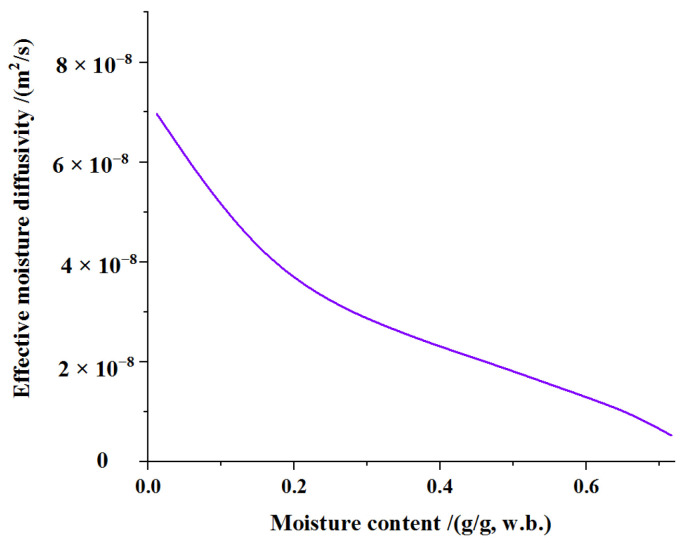
Dependence of effective moisture diffusivity on moisture content.

**Figure 5 molecules-27-04421-f005:**
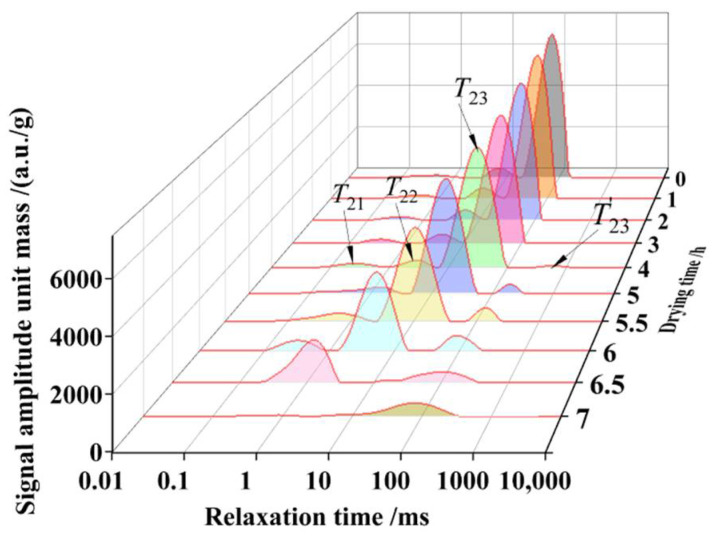
LF-NMR transverse relaxation time spectra of soybean dregs with different moisture contents. *T*_21_ is bound water; *T*_22_ is immobilized water; *T*_23_ and *T*_23_’ are free water.

**Figure 6 molecules-27-04421-f006:**
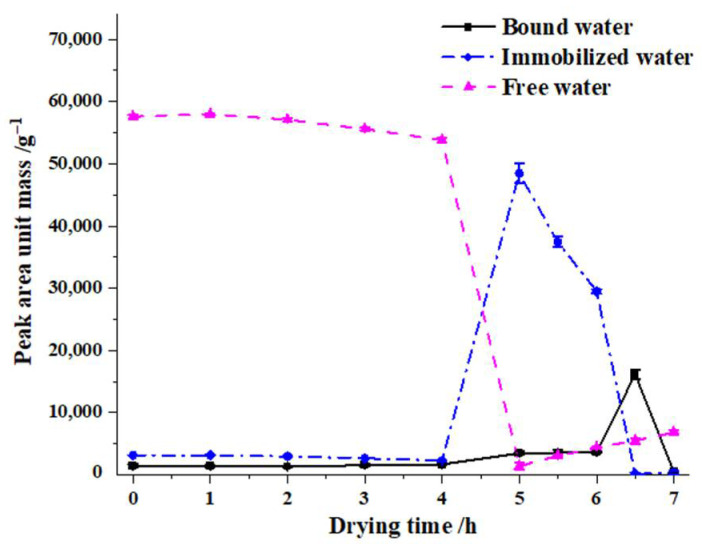
Moisture status of soybean dregs at different drying stages.

**Figure 7 molecules-27-04421-f007:**
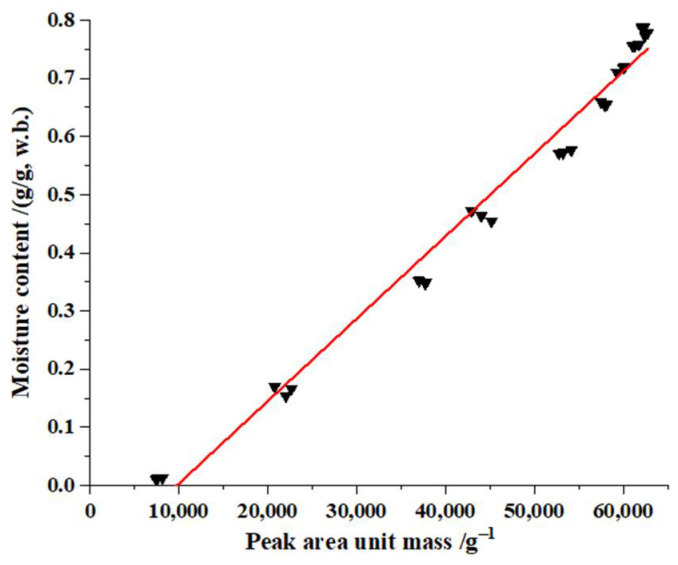
Fitting curve between the peak area unit mass and moisture content.

**Figure 8 molecules-27-04421-f008:**
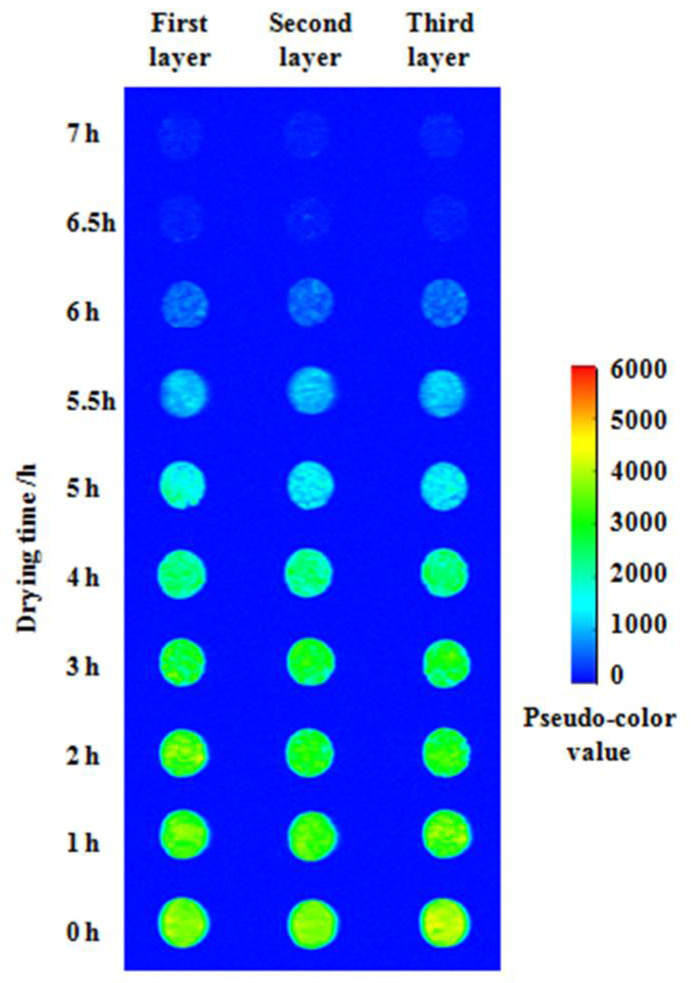
Pseudo-color picture of the soybean dregs during the drying process.

**Figure 9 molecules-27-04421-f009:**
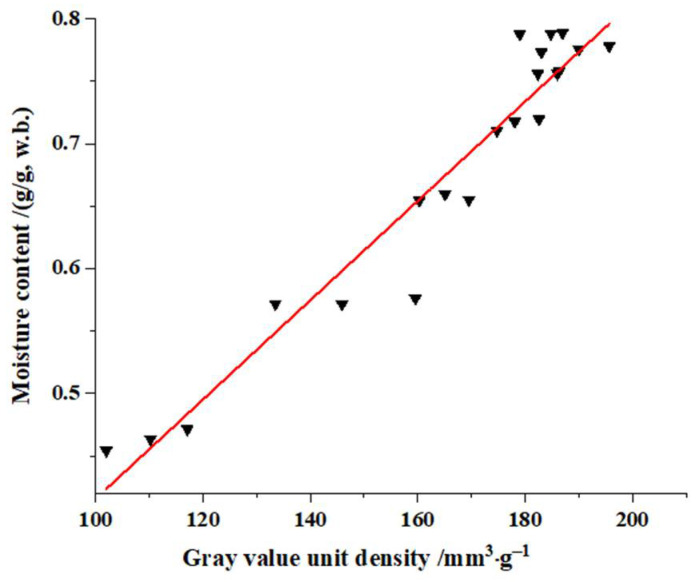
Fitting curve between the image gray value of the LF-NMR and the moisture content.

**Table 1 molecules-27-04421-t001:** Relaxation peak times of soybean dregs with different moisture contents.

Drying Time/h	T_21_/ms	T_22_/ms	T_23_/ms	T_23_’/ms
0	0.48 ± 0.04 ^b^	7.06 ± 0.00 ^c^	69.08 ± 5.69 ^bc^	-
1	0.41 ± 0.03 ^b^	6.44 ± 0.53 ^c^	65.79 ± 0.00 ^bc^	-
2	0.40 ± 0.03 ^b^	5.34 ± 0.00 ^cd^	57.22 ± 0.00 ^bc^	-
3	0.31 ± 0.02 ^b^	4.04 ± 0.00 ^de^	43.29 ± 0.00 ^c^	-
4	0.25 ± 0.00 ^b^	2.79 ± 0.23 ^e^	29.90 ± 0.2.46 ^c^	487.33 ± 40.13
5	1.09 ± 0.58 ^a^	17.11 ± 1.41 ^a^	288.67 ± 151.98 ^a^	-
5.5	0.60 ± 0.09 ^b^	10.72 ± 0.00 ^b^	138.79 ± 11.43 ^b^	-
6	0.34 ± 0.03 ^b^	5.34 ± 0.00 ^cd^	91.32 ± 7.52 ^bc^	938.57 ± 130.67
6.5	1.26 ± 0.10 ^a^	16.3 ± 0.00 ^a^	100.00 ± 0.00 ^bc^	-
7	0.57 ± 0.35 ^b^	5.47 ± 2.74 ^cd^	79.42 ± 6.54 ^bc^	-

Different letters in the same column indicate a significant difference between the means (*p* < 0.05).

**Table 2 molecules-27-04421-t002:** Comparison of the predicted and measured moisture content of soybean dregs.

Drying Time/h	Measured Moisture Content/(g/g, w.b.)	*T*_2_ Relaxation Signal			Hydrogen Proton Imaging		
		Peak Area Unit Mass/g^−1^	Predicted Moisture Content/(g/g, w.b.)	Predicted Error/%	Measured Gray Value	Predicted Moisture Content/(g/g, w.b.)	Predicted Error/%
2.5	0.7435	61,327.98 ± 270.31	0.7309	1.69	179.64 ± 5.78	0.7332	1.39
4.5	0.6206	55,219.64 ± 343.20	0.6441	3.79	156.72 ± 12.77	0.6422	3.48
6.6	0.1317	19,930.19 ± 737.15	0.1430	8.58	-	-	-

## Data Availability

Not applicable.

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
