# Peer review of "Water Status and Predictive Models of Moisture Content during Drying of Soybean Dregs Based on LF-NMR"

_molecules, 2022, doi:10.3390/molecules27144421_

Round 1

Reviewer 1 Report

Comments attached.

Reviewer 2 Report

COMMENTS

 ABSTRACT SECTION:

The abstract exceeds the maximum number of words (200 word). Please check the instructions for authors.

 KEYWORDS

Lines 30-31. Use words other than those in the title. This increases the chances of the article being found, read, and cited. Some suggestions: by-products, processing, relaxation peak.

RESULTS AND DISCUSSION SECTION:

Figure 1. There are no error bars. Only one sample was measured?

Figure 4. There is an inconsistency in the data. Peak area unit mass does not correspond to the values reported in Figure 3. In addition, include all measurements (triplicate) for fitting curve (30 points).

Please also check table 1.

Figure 6. Please include all measurements.

Please improve the results and discussion in the manuscript.

Reviewer 3 Report

The present work deals with the drying of bean greg based on low-field nuclear magnetic resonance and determination of water migration rules. The work is well written, but I believe it is possible to improve it before publishing it. My suggestions can be checked below.

1. Scientific article writing is better when it is impersonal.

2. It would be interesting if the authors cited works that used the LF-NMR technique in drying studies. It would be possible?

3. Why did the authors only use the temperature of 80 °C in the tests? Why did you choose this value? What were the preliminary studies reported by the authors?

4. As the research is well focused on drying, it would be better to provide more details on the experimental drying apparatus.

5. Equation 1 needs to be further explained. If the authors are using md as the dry mass obtained at the drying temperature of 80 °C, what would be the initial mass of the sample and the mass of the sample that varies with time? Please explain.

6. A picture of the material studied would be interesting.

7. As the authors discuss, moisture removal is slower at the beginning of drying and faster at the end. This contradicts the classical drying processes, since, in the final moments of drying, the removal of moisture is slower due to a greater resistance of the diffusive mechanisms of moisture transport. Why the results presented in Fig. 1 differ so much from traditional results? How different is drying using the LF-NMR technique?

8. In Figure 3, it is shown that, at the beginning of the process, there is a large amount of free water until a time of approximately 4 h. When there is free water, the evaporation rate is higher and drying, consequently, occurs more quickly. Compared to figure 1, wouldn't that be contradictory? The fact that the solid has bound moisture, especially at the end of the process, would indicate slower drying, as the removal of moisture is more difficult in this condition. I would expect that due to the presence of free moisture at the beginning of drying and bound moisture at the end, the results would be reversed.

9. I think it would be more interesting if the authors indicated the advantages and disadvantages of LF-NMR in drying studies. This could be placed at the conclusion of the work.

Reviewer 4 Report

The manuscript "Water migration rules and predictive models during bean dreg drying based on low-field nuclear magnetic resonance" is about the water migration during bean dreg drying. However, there are some problems in the paper needed to be clarified, which are shown in the following comments

Point 1: The introduction should be given about drying methods of bean dreg more detailed. The authors should indicate the advantages and limitations of methods studied in the literature. Explain the pros and cons of the drying methods.

Point 2: The percentage of references in Chinese with English abstract is 88.6 %. I would suggest that more studies in English should be included in the reference. 

Point 3: The information of effective diffusion coefficient in porous media is very important. Please calculate the diffusion coefficient in the study.

Point 4: The quality of the graphs is not good. For example, Figure 1, 4 and 6 are not clear and can be improved.

Point 5: In Figure 3, what are the meanings of “Combined water” and “Semi-combined water? Please illustrate them and revise.

Point 6: In results and discussion section, the authors should probe into their results with other approach for bean drying. To contribute more to this field, the authors should compare their results with those in relevant published works.

Point 7: Word abbreviation must be illustrated. For example, line 101 CPMG?  line 110 MRI?

Point 8: Line 203, what is the meaning of “established equation (2)”?

Point 9: An expression is not consistent in the article, for example, percentage points or % (line 131-133 and line 276-278).

Point 10: Line 98 and line 337, there are a number of typos, please correct.

Date of this review

15 June 2022

Round 2

Reviewer 1 Report

The authors have carefully addressed all my comments. The quality of the manuscript has been significantly improved and is acceptable for publication.

Reviewer 2 Report

Dear authors, thank you for accepting the suggestions and making the changes. The manuscript has improved.

Reviewer 3 Report

I accept the response of the authors for my comments. I only suggest authors to indicate more literature references to my comments n° 6 and 7 that support their discussion.

Reviewer 4 Report

The manuscript " Water Status and Predictive Models of Moisture Content during Drying of Soybean Dregs Based on LF-NMR " is about the water status and predictive approach during bean dreg drying.

There is a point need to be revised. Please find my comment below:

Point 1: Line 113, the term of right-hand side in Eq. 2 is a typing mistake, please correct.

Date of this review

5 July 2022